# Identification of Differentially Expressed miRNAs in Porcine Adipose Tissues and Evaluation of Their Effects on Feed Efficiency

**DOI:** 10.3390/genes13122406

**Published:** 2022-12-19

**Authors:** Mingxing Liao, Zhuqing Ren, Yuanxin Miao

**Affiliations:** 1Key Laboratory of Agriculture Animal Genetics, Breeding and Reproduction of the Ministry of Education & Key Laboratory of Swine Genetics and Breeding of the Ministry of Agriculture and Rural Affairs, College of Animal Science, Huazhong Agricultural University, Wuhan 430070, China; 2Research Institute of Agricultural Biotechnology, Jingchu University of Technology, Jingmen 448000, China

**Keywords:** residual feed intake, feed efficiency, microRNA, pig, adipose tissues

## Abstract

Feed efficiency (FE) is a very important trait affecting the economic benefits of pig breeding enterprises. Adipose tissue can modulate a variety of processes such as feed intake, energy metabolism and systemic physiological processes. However, the mechanism by which microRNAs (miRNAs) in adipose tissues regulate FE remains largely unknown. Therefore, this study aimed to screen potential miRNAs related to FE through miRNA sequencing. The miRNA profiles in porcine adipose tissues were obtained and 14 miRNAs were identified differentially expressed in adipose tissues of pigs with extreme differences in FE, of which 9 were down-regulated and 5 were up-regulated. GO and KEGG analyses indicated that these miRNAs were significantly related to lipid metabolism and these miRNAs modulated FE by regulating lipid metabolism. Subsequently, quantitative reverse transcription–polymerase chain reaction (qRT-PCR) of five randomly selected DEMs was used to verify the reliability of miRNA-seq data. Furthermore, 39 differentially expressed target genes of these DEMs were obtained, and DEMs–target mRNA interaction networks were constructed. In addition, the most significantly down-regulated miRNAs, ssc-miR-122-5p and ssc-miR-192, might be the key miRNAs for FE. Our results reveal the mechanism by which adipose miRNAs regulate feed efficiency in pigs. This study provides a theoretical basis for the further study of swine feed efficiency improvement.

## 1. Introduction

Both inputs and outputs are the main factors affecting pig producers’ profit, and input cost reduction can improve the economic profits of pig-breeding enterprises [1]. Feed cost is a major input cost, accounting for about two-thirds of the total cost of pig breeding [2]. Feed efficiency (FE) is defined as the ratio of input (feed intake, FI) to output (production), and thus FE improvement can decrease feed cost in pig enterprises. Residual feed intake (RFI) is one of the most common measurements for FE, and RFI is defined as the difference between the actual FI and the predicted FI calculated based on the animal’s body size and growth rate over a period of time [3,4,5]. Since a low RFI means high FE and low FI, RFI can be used as a negative selection trait to improve FE [6,7,8]. Additionally, the exploration of the molecular mechanism of RFI will be helpful in the improvement of feed efficiency.

Continuous selection of low RFI in pigs for multiple generations can effectively improve the feed efficiency of pigs. It has been reported that low-RFI pigs exhibit similar growth rate but lower FI and feed conversion ratio (FCR) compared to high-RFI pigs [9,10]. Nutrient utilization and metabolism in the tissues of low-RFI pigs differ from those of high-RFI pigs. What’s more, low-RFI pigs have lower oxidase activities and lower glycolytic capacity in skeletal muscles [9,11]. Low-RFI pigs have been reported to reduce the oxidation of nutrients to generate more ATP which is subsequently released as heat [12,13,14]. Lower antioxidant protein expression and lower ROS (reactive oxygen species) production have been reported in the longissimus muscle mitochondria of low-RFI pigs [15,16]. Some studies of the physiological processes of individuals with extreme RFI have explained part of the variation mechanism of RFI, but the mechanism of FE remains to be further investigated.

miRNAs are noncoding RNA molecules of roughly 19 to 25 nt in length, and they can regulate the expression levels of gene by binding to specific messenger RNA (mRNA) sequences [17,18]. In domestic animals, miRNAs have been reported to play a pivotal part in regulating skeletal muscle and fat development as well as immune response [19,20,21]. miRNAs are key regulators of FE. In Nelore cattle, one miRNA in skeletal muscle and four miRNAs in liver have been found to regulate feed-efficiency-related biological pathways [22]. In the livers of cattle, 49 DEMs have been identified to be related to RFI, of which 33 down-regulated DEMs in low-RFI cattle played important roles in physiological pathways associated with FE [23]. A total of 15 DEMs were identified in skeletal muscle between high- and low-RFI pigs, and they mainly take part in skeletal muscle growth and development [24]. In addition, 14 DEMs have been found in the liver of pigs with significant RFI differences, and they are mainly related to the signaling pathways of insulin, GnRH and mTOR [25]. These results indicate that miRNAs are vital regulators of FE.

Adipose tissue is a major sensor that modulates various processes such as FI, energy homeostasis, fat metabolism and whole-body physiological processes [26,27]. Lipid metabolism in adipose tissue has been reported to play an important role in regulating the FE of pigs [28,29,30,31]. Fatness traits, including backfat thickness and intramuscular fat content, were weakly negatively correlated with FE, so could not be used as an indicator to improve growth rate [32,33]. In this study, we used miRNA sequencing to identify multiple DEMs from the adipose tissues of pigs with significantly different RFIs. GO and KEGG pathway analyses of target genes of the DEMs were performed. Subsequently, we analyzed the interactions between DEMs identified in this study and our previously reported DEGs and visualized them. Our findings provide a theoretical basis for the further study of swine-feed-efficiency improvement.

## 2. Materials and Methods

### 2.1. Sample Preparation and RNA Isolation

This study selected six castrated boars with no difference in body weight and extreme differences in FE based on the RFI value (3 high RFI versus 3 low RFI) for the experiments (Appendix A). The individuals were from populations of Yorkshire pigs (*n* = 236), and the feed intake was measured by ACEMA64 (ACEMO, Pontivy, France) automated individual feeding systems [24]. The RFI values of the population were calculated and the distribution of RFI (−0.03188 ± 0.2051) in the population is shown in Figure 1. The performances of individuals with extreme RFI differences (*n* = 50) were compared (Table 1) and the results showed that FI and fat deposition were significantly reduced in low-RFI (high-FE) pigs, which is compatible with the results reported in previous studies [34,35]. The experiments in which animals used for miRNA-sequencing were raised, slaughtered and sampled were approved by the Animal Care and Use Ethics Committee of Huazhong Agricultural University (permit number: HZAUMU2013-0005). According to the results of RFI, dorsal subcutaneous adipose tissue samples comprising all fat layers were sampled from 6 animals with extreme RFI differences at the last rib level and immediately immersed in liquid nitrogen within 30 min, and subsequently transferred to −80 °C for storage. For total RNA extraction, all samples of the frozen subcutaneous adipose tissue were extracted using TRIzol reagent (Invitrogen, Carlsbad, CA, USA). The RNA integrity and purity were examined using a Bioanalyzer 2100 (Agilent Technologies, Santa Clara, CA, USA) and a NanoDrop 2000 spectrophotometer (Thermo Scientific, Waltham, MA, USA).

### 2.2. Small RNA Library Construction and Sequencing

Six miRNA-sequencing libraries were constructed from 3 high-RFI pigs and 3 low-RFI pigs by TruSeq^®^ Small RNA library Kit (ILLumina^®^, San Diego, CA, USA). The obtained miRNA-sequencing libraries were purified with AMPure XP system, and the quality was assessed on the Agilent 2100 Bioanalyzer (Agilent Technologies, Santa Clara, CA, USA). After quality control, the high-quality miRNA-sequencing libraries were sequenced on Illumina HiSeq3000 platform (Illumina, San Diego, CA, USA).

### 2.3. Analysis of miRNA Sequencing Data

The raw reads obtained by miRNA sequencing were processed with Trimmomatic (v0.39) to trim adapter contaminants and remove substandard reads. Subsequently, the clean reads were aligned to the pig reference genome with miRdeep2 [36]. In pigs, known pig miRNAs were identified by aligning all clean reads to the pig miRNA reference sequences in miRBase database (version 22). The sequences matching mature miRNAs in miRBase database and aligned to pig reference genome were identified as known miRNAs reads. The sequences aligned to the pig reference genome and matching miRNA databases were considered as potential miRNA reads. MiRDeep (v2.0.0.7) was used to predict novel miRNAs for those sequences unmatched to known pig miRNAs [37]. The secondary structures of novel miRNA were plotted using RNAfold software.

### 2.4. Identification of DEMs and Their qRT-PCR Validation

The miRNA expression level was normalized by transcript per million (TPM). The DEMs between the high-FE and low-FE pigs were analyzed with the DESeq R package (v4.0.3). The thresholds to determine the DEMs were set as following: *p*-value less than 0.05 and absolute value of log2 (Fold change) greater than or equal to 1.

In order to check the reliability of miRNA sequencing, 5 adipose samples from high FE pigs and 5 from low FE pigs were selected to quantify the relative expression of the DEMs by qRT-PCR. A total of 5 miRNAs were randomly selected for qRT-PCR validation; the primer sequences of miRNAs are listed in Appendix A. The miRNA was reverse transcribed into cDNA. The cDNA was reverse transcribed with 1 μg of total RNA using the Mir-X miRNA First-Strand Synthesis Kit (TaKaRa, Tokyo, Japan) according to the manufacturer’s instructions. The qRT-PCR was performed following standard protocols on Roche Lightcycler 480 Real-Time PCR System with SYBR Green PCR Master Mix (TOYOBO, QPK201) and porcine U6 snRNA was used as an internal control for miRNA. Reactions were performed thrice and each well contained 1 μL cDNA, 5 μL 2 × SYBR Green PCR Master Mixture, 0.1 μL each primers and 3.6 μL RNase-free water. The reaction conditions were preincubated at 95 °C for 5 min and 40 PCR amplification cycles of 95 °C for 30 s, 60 °C for 30 s and 72 °C for 15 s; at the end of amplification process dissociation curves were generated to validate the data quality. The relative expression levels of miRNAs were calculated using the 2^−ΔΔCt^ method and Student’s *t*-test was used to analyze the expression difference between the high-FE and low-FE pigs.

### 2.5. Prediction of miRNA Target Gene, Gene Ontology (GO) and KEGG Pathway Enrichment Analyses

DIANA miRPath (v.3), an online miRNA target-prediction tool, predicted miRNA targets in CDS or 3′-UTR regions, and homologous human miRNAs were used to predict the potential target genes of DEMs. Go and KEGG pathway-enrichment analyses of target genes of miRNAs were performed using the DAVID Bioinformatics Resources. The overlap between differentially expressed genes which were identified in our previous paper [38] and potential target genes were considered as the differentially expressed target genes. The network of 12 differentially expressed miRNAs and 39 differentially expressed target genes in high- and low-FE pigs was plotted using Cytoscape v3.6.1.

## 3. Results

### 3.1. Characterization of miRNA-Sequencing Data

In order to identify DEMs in the adipose tissue between the high-FE group and the low-FE group, three samples per group were sequenced using Solexa sequencings (Appendix A). A total of 12.83~26.77 million raw reads were generated from each sample. After excluding short reads and low-quality adaptor sequences, 10.61~20.01 million clean reads were retained for each sample, accounting for 82.7~91.2% of the raw reads per sample (Table 2). The clean reads length mainly ranged from 21 to 23 nt with 22 nt exhibiting the peak, indicating that sRNAs were mainly miRNAs (Appendix A).

### 3.2. Differentially Expressed miRNAs between High-FE and Low-FE Pigs

To reveal the role of miRNAs in regulating FE in adipose tissues, we identified differentially expressed miRNAs (DEMs) in adipose tissues between high-FE and low-FE pigs. A total of 14 DEMs were identified, of which 5 miRNAs (ssc-miR-582-5p, ssc-miR-150, ssc-miR-155-5p, ssc-miR-331-5p and ssc-miR-196a) were up-regulated and 9 miRNAs (ssc-miR-129a-5p, ssc-miR-9, ssc-miR-138, ssc-miR-194b-5p, ssc-miR-194a-5p, ssc-miR-192, ssc-miR-10386, ssc-miR-136-5p and ssc-miR-122-5p) were down-regulated (Figure 2, Table 3).

Of these 14 DEMs, 13 were homologous to human miRNAs and 1 miRNA (ssc-miR-10386) was not homologous to human miRNAs (Figure 3). Cluster analysis showed that the high-FE group was separated from the low-FE group, and that the expression patterns of miRNAs exhibited significant differences between the high-FE group and the low-FE group.

### 3.3. Verification of miRNA Sequencing Data by qRT-PCR

We randomly selected five DEMs (ssc-miR-122-5p, ssc-miR-192, ssc-miR-155-5p, ssc-miR-150 and ssc-miR-9) for qRT-PCR so as to validate the accuracy of miRNA sequencing data. The results revealed that the expression levels of ssc-miR-155-5p and ssc-miR-150 were higher in the high-FE group than in the low-FE group; nevertheless, the expression levels of ssc-miR-122-5p, ssc-miR-192 and ssc-miR-9 were lower in the high-FE group than in the low-FE group. The results of the five randomly selected miRNAs showed that the results of qRT-PCR were compatible with the results of miRNA sequencing, suggesting the miRNA sequencing data is reliable (Figure 4).

### 3.4. Target-Gene Prediction of DEMs

In animals, miRNA can mediate the level of the post-transcriptional genes by complementing the 2nd to 7th nucleotides of the 3′UTR [39,40]. In order to investigate the role of DEMs in the adipose tissues of pigs with extreme FE differences, homologous human miRNAs of pig DEMs were employed to predict the target genes. As a result, 8962 target genes were obtained from 12 homologous human miRNAs of pig DEMs, and 5927 target genes were retained after removing the duplicates (Appendix A).

### 3.5. Functional Enrichment Analysis of DEMs in Adipose Tissues

To investigate the biological functions of DEMs, the target genes of DEMs were enriched by GO and analyzed by KEGG pathway. The GO enrichment analysis identified 1034 significantly enriched biological process GO terms, 288 significantly enriched cellular component GO terms and 259 significantly enriched molecular function GO terms (Appendix A). GO enrichment analysis showed that the top 20 biological processes in which the target genes were significantly enriched mainly included positive/negative regulation of transcription from RNA polymerase II promoter, positive/negative regulation of transcription (DNA-templated), cell division, cellular response to DNA damage stimulus, cell cycle and protein phosphorylation. Cellular components in which most target genes were significantly enriched mainly consisted of nucleoplasm, nucleus, cytosol, membrane and cytoplasm. Molecular functions in which the target genes were significantly enriched mainly involved protein binding, chromatin binding, cadherin binding and metal-ion binding. The top 20 significant GO enrichments are shown in Figure 5.

KEGG analysis results showed that 57 pathways were significantly enriched with the target genes of DEMs (Appendix A), of which the top 20 are shown in Figure 6. The significantly enriched KEGG pathways mainly involved the AMPK signaling pathway, ubiquitin-mediated proteolysis, insulin resistance, Hippo signaling pathway, FoxO signaling pathway, insulin signaling pathway and p53 signaling pathway. The hierarchical clustering analysis results of the relationships between DEMs and their target gene pathways showed that functionally similar miRNAs were clustered together (Figure 7). The miRNAs with similar functional categories and the same regulation pattern were clustered together, for example miR-129-5p and miR-196a-5p were clustered together. The miR-129-5p and miR-196a-5p have similar functions and are often reported simultaneously [41,42].

### 3.6. miRNA–mRNA Association Analysis

In order to understand the molecular mechanism of FE, we investigated a large number of DEGs from adipose tissues of high-FE and low-FE pigs [38]. By integrated analysis of 147 of our previously reported DEGs (Appendix A) and 12 DEMs obtained in this study, we obtained 39 candidate target genes from 12 miRNAs, and all these 39 target genes were differentially expressed in adipose tissues between the high-FE group and the low-FE group (Figure 8).

## 4. Discussion

As a vital trait, FE strongly affects feed cost and economic benefits in pig-breeding enterprises. Selection of pigs with high FE can effectively reduce feed intake so as to save feed costs [34]. Previous studies have shown that energy metabolism contributes to the FE of pigs [24,43]. Adipose tissue is one of the most crucial tissues that take part in energy metabolism and is associated with the FE of pigs [38]. Furthermore, adipose tissue can secrete many adipocytokines to mediate appetite, energy homeostasis, and lipid and glucose metabolism [44]. Adipose tissue growth, extracellular matrix formation, lipid metabolism, inflammatory response and immune response have been reported related to FE in pigs [45]. Moreover, the phenotypes of high-FE pigs were compared with those of the low-FE pigs; it was found that high-FE pigs have thinner backfat depth and lower fat content [11,34,46]. In our study, the backfat thickness was also decreased in high-FE pigs; although the decrease was small, the difference was significant. Lipogenesis and fat partitioning in adipose tissue have been reported to be related to energetic efficiency [47,48]. In addition, it has been reported that subcutaneous adipose tissue thickness contributes to 2~5% of variation in feed intake [49]. Therefore, adipose tissue acts as an energy depot to mediate metabolic homeostasis and nutrient availability, thereby further regulating feed efficiency [50]. miRNAs can modulate various biological processes through modulating tissue that takes part in energy metabolism and is associated with pig feed efficiency [24,25,51,52,53]. In this study, the miRNA expression profiles of adipose tissues from high-FE and low-FE pigs were compared to reveal the miRNA-mediated regulation of FE. We identified 14 DEMs in the adipose tissues of high-FE and low-FE pigs, and the identification of miRNAs, target genes and the concerning important signaling pathways will be useful in the development of strategies for improving FE.

KEGG pathway analyses of target genes of the DEMs contribute to understanding the regulatory mechanism of DEMs. The significantly enriched KEGG pathways are involved in the AMPK signaling pathway, mTOR signaling pathway, ubiquitin-mediated proteolysis, Hippo signaling pathway, insulin signaling pathway and adipocytokine signaling pathway. In the hypothalamus, the mTOR signaling pathway has been identified to be associated with FE, and the genes related to this pathway are down-regulated in high-FE pigs [54]. RNA sequencing in the muscle of the Pacific white shrimp showed that the PI3K-Akt signaling pathway, AMPK signaling pathway and mTOR signaling pathway mediated the level of the genes associated with FE [55]. The Hippo signaling pathway, insulin signaling pathway and adipocytokine signaling pathway are important pathways related to FE [56,57,58]. Thus, miRNAs can affect these pathways through mediating the level of the target genes.

Of the 14 DEMs, 9 were down-regulated in the adipose tissues in high-FE pigs, relative to low-FE pigs, with the expressions of ssc-miR-122-5p and ssc-miR-192 exhibiting the maximum down-regulation. Previous studies have demonstrated that ssc-miR-122-5p and ssc-miR-192 participate in fat metabolism [59,60,61,62,63]. ssc-miR-122 is a key regulator of lipid metabolism [64,65], and our previous study has revealed that *B4GALT6* (a predicted target gene of ssc-miR-122) is down-regulated in the adipose tissue of high-FE pigs, in contrast to that of low-FE pigs [38]. *B4GALT6* has been reported to take part in the lipid biosynthetic process of adipose tissue [66]. It is worth noting that ssc-mir-122 has been found to be a candidate miRNA for average daily gain (ADG) and days (AGE) traits of pigs [67]. As a key miRNA for lipogenesis, ssc-miR-192 can regulate adipose deposition and differentiation [68,69], and it can also target the *B4GALT6* gene to regulate lipid metabolism in adipose tissue. Our KEGG pathway analysis showed that the target genes of these two miRNAs were mainly enriched in the AMPK signaling pathway, Wnt signaling pathway, fatty acid metabolism, insulin resistance, insulin signaling pathway and TGF-β signaling pathway. Thus, ssc-miR-122 and ssc-miR-192 in the porcine adipose tissues might modulate the FE by regulating lipid metabolism.

A previous study has revealed that ssc-miR-194a-5p is involved in lipid and cholesterol metabolism through the target genes *Apoa5* and *Hmgcs2* [70,71], and our data showed that this miRNA was significantly reduced in the adipose tissues of high-FE pigs. *ELOVL7*, a predicted target gene of miR-194, has been found to be involved in the lipid biosynthetic processes of adipose tissues and the synthesis of FE-related polyunsaturated fatty acids (PUFAs) in pigs, and our previous study has indicated down-regulation in the adipose tissue of high-FE pigs, relative to that of low-FE pigs [38,72,73,74,75]. miR-9, a fat-formation-related biomarker, was also predicted to target *ELOVL7* and was down-regulated in the adipose tissues of high-FE pigs. It has been reported that miR-9 can induce the differentiation of preadipocytes [76]. In pigs, miR-9 is highly expressed in obese pigs, and it contributes to lipid accumulation [77].

Our data showed that miR-155, a brown adipogenesis inhibitor, was up-regulated in the adipose tissue of high-FE pigs, compared with that of low-FE pigs [78,79]. Previous studies have shown that miR-155 knockout can produce more heat and increase insulin sensitivity, and that overexpressing miR-155 in mice can reduce brown adipose tissue mass and the level of thermogenic markers [79]. In pigs, miR-155 is highly expressed in the fat tissues, and it is a key positive regulator of the TLR3/TLR4 signaling pathway [80]. Our data showed that miR-150 was up-regulated in the adipose tissues of high-FE pigs, and this miRNA has been reported to modulate adipose tissue function [81]. Overexpression of miR-150 can facilitate the proliferation of adipocytes, suppress adipocyte differentiation and decrease the formation of lipid droplets [82]. miR-138, an inhibitor of adipogenesis, is down-regulated in the adipose tissues of high-FE pigs, in contrast to that of low-FE pigs [83]. It has been reported that overexpression of miR-138 can inhibit lipid-droplet accumulation [84]. miR-129 and miR-196a have similar functions. It has been reported that miR-129 is involved in regulating lipid accumulation and thermogenesis [85,86], and our data showed that miR-129 was significantly reduced in the adipose tissues of high-FE pigs. Previous studies have shown that overexpression of miR-129 can inhibit adipogenesis and inhibiting the expression of miR-129 can promote adipogenic differentiation [87,88]. miR-196a is a key regulator of fat deposition [89,90], and our data showed that miR-196a is up-regulated in the adipose tissue of high-FE pigs, in contrast to that of low-FE pigs. In pigs, miR-196a has been reported to be associated with preadipocyte differentiation and adipogenesis [91,92]. Overexpression of miR-196a can increase lipid accumulation and promote preadipocyte differentiation [93]. miR-331, a microRNA participating in regulating the proliferation, differentiation and fatty-acid accumulation of porcine preadipocytes, was up-regulated in the adipose tissues of high-FE pigs [94]. Overexpressing miR-331 can increased fatty-acid synthesis [95]. All of these miRNAs can regulate energy metabolism and adipocytokine secretion by influencing lipogenesis and fat partitioning in adipose tissue, thereby affecting feed efficiency.

The miRNA–mRNA association analysis revealed 39 candidate target genes from 12 miRNAs. Adiponectin, an endocrine factor secreted by adipose tissue, is involved in regulating food intake and energy expenditure [96,97]. *ALDH1A3* and *CYP3A5*, the DEGs predicted as miR-9 and miR-122-5p targets, participate in adiponectin expression through regulating retinoic acid metabolic [98,99,100]. Ca^2+^ can mediate the feed efficiency by the cAMP signaling pathway [38]. Suppression of *CLIC4*, the DEGs predicted as miR-122-5p, miR-155-5p, miR-196a and miR-9 targets, can significantly enhanced Ca(2+) release [101]. *GABRE*, the DEGs predicted as miR-122-5p, are involved in GABA signaling and mediate Ca^2+^ signals [102]. It has been reported that ATP synthesis is associated with FE [103]; the genes *OAS1*, *MET*, *RHOBTB3*, *HSPA4L*, *KIF20A*, *MST1R* and *MKI67* are related to ATP binding and there were DEGs predicted as DEM targets. Therefore, DEMs may affect the FE of pigs by targeting the DEGs involved in lipid metabolism and energy metabolism.

## 5. Conclusions

In conclusion, a total of 14 significant DEMs were identified from the adipose tissues of high-FE pigs and low-FE pigs, of which 13 miRNAs were homologous to human miRNAs. The functional analysis of miRNAs and target genes showed that these DEMs modulated FE by regulating lipid metabolism, adiponectin, energy metabolism and appetite. The DEMs mediated the level of adipocytokine and fatty-acid accumulation by targeting the DEGs; subsequently, adipocytokines were secreted by the adipose tissue to regulate systemic metabolism to affect feed efficiency. Taken together, our findings give a new insight into the molecular mechanisms of miRNAs in regulating pig feed efficiency.

## Figures and Tables

**Figure 1 genes-13-02406-f001:**
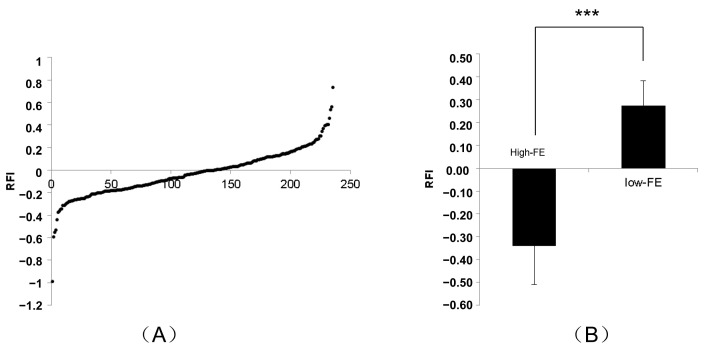
Summary of RFI in the populations of Yorkshire pigs. (**A**) The distribution of RFI in the populations of Yorkshire pigs. (**B**) The RFI of the sequencing individual. *** represent *p* ≤ 0.001, meaning the RFI between High-FE and Low-FE significant differences.

**Figure 2 genes-13-02406-f002:**
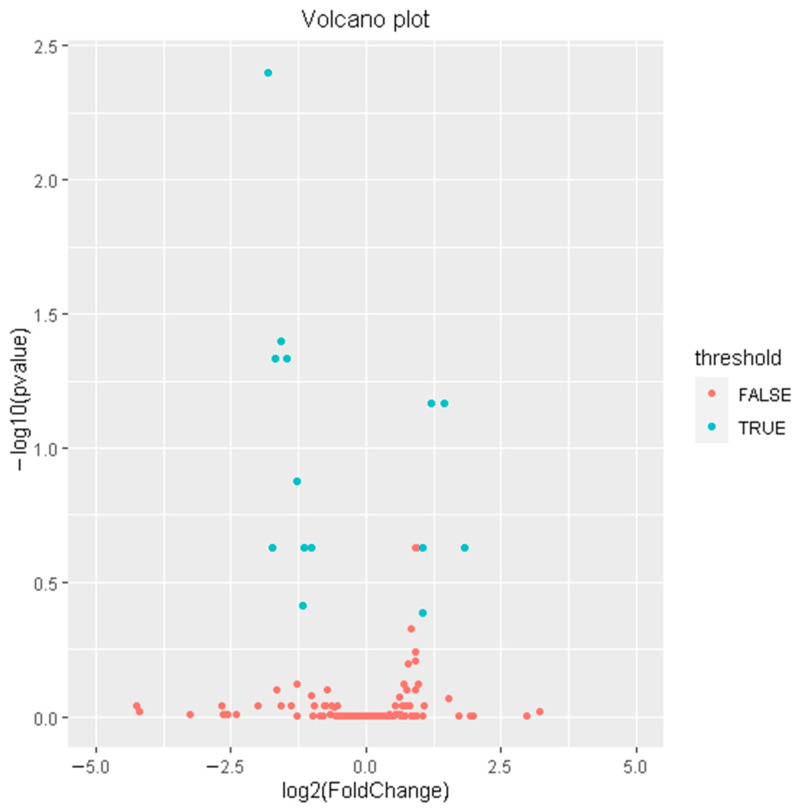
Volcano plot of DEMs in adipose tissues between high-FE and low-FE pigs. The blue dots represent the DEMs; the red dots represent the miRNAs with no significant expression differences.

**Figure 3 genes-13-02406-f003:**
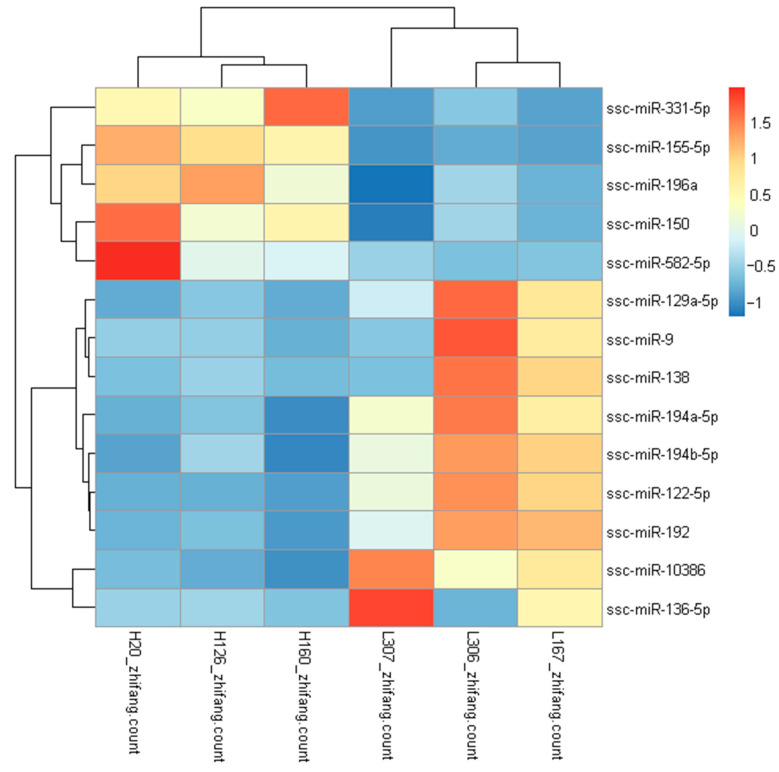
Hierarchically cluster heat map of DEMs. Red represents up-DEMs in adipose tissue, and blue represents down-DEMs in adipose tissue. Color intensity represents the degree of regulation (up- or down-regulation).

**Figure 4 genes-13-02406-f004:**
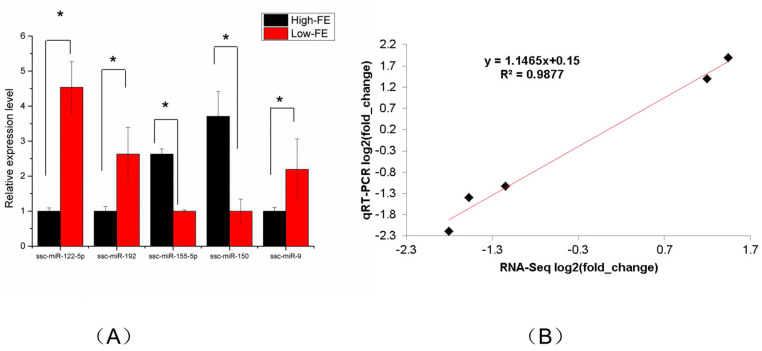
Validation of the RNA sequencing using qRT-PCR. U6 snRNA was used for samples standardization. (**A**) qRT-PCR validation of five DEMs in adipose tissues. (**B**) Line-fit plot of qRT-PCR results and RNA-sequencing data for selected DEMs. Mark * in (**A**) represent the expression level significantly different between two group (*p* < 0.05).

**Figure 5 genes-13-02406-f005:**
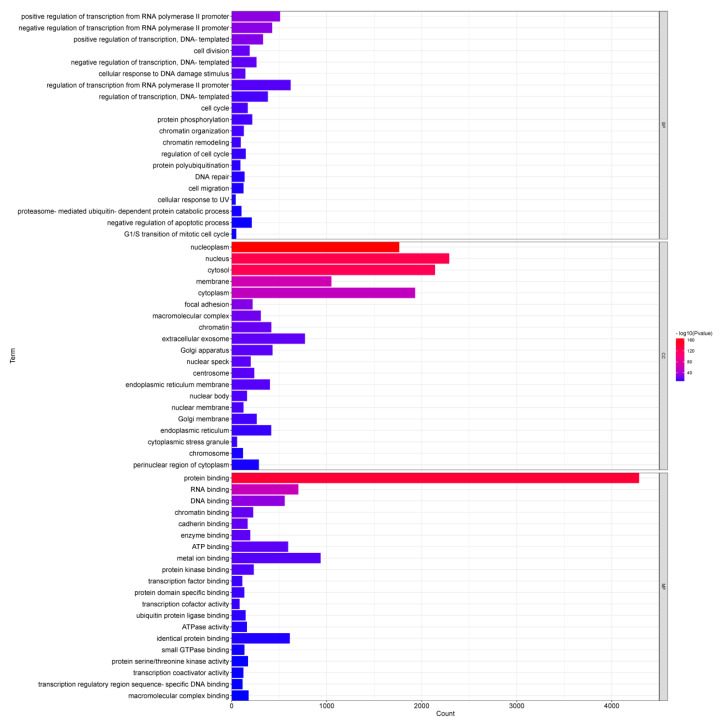
GO enrichment analysis of target genes of DEMs.

**Figure 6 genes-13-02406-f006:**
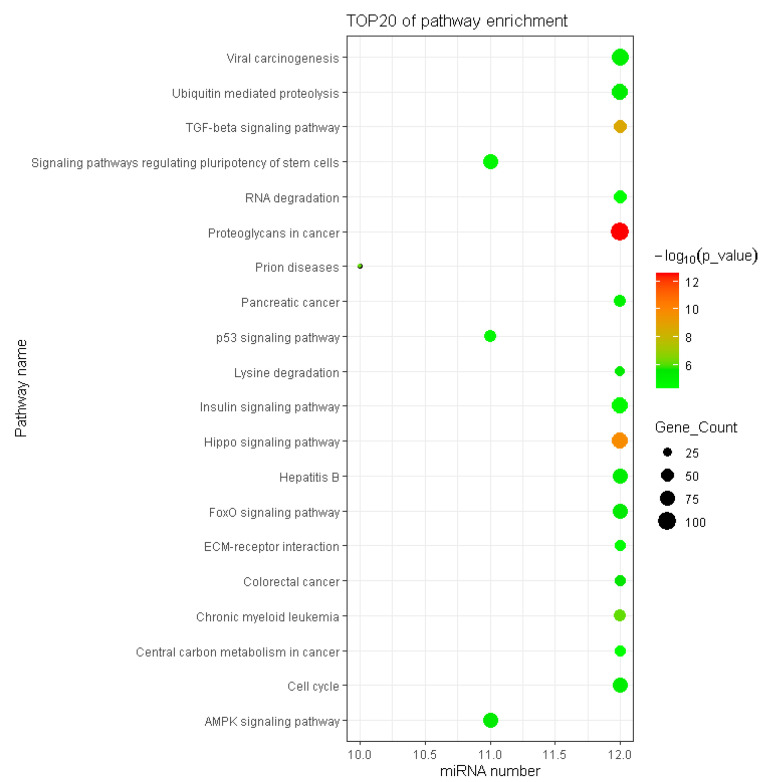
Pathway enrichment of target genes of DEMs.

**Figure 7 genes-13-02406-f007:**
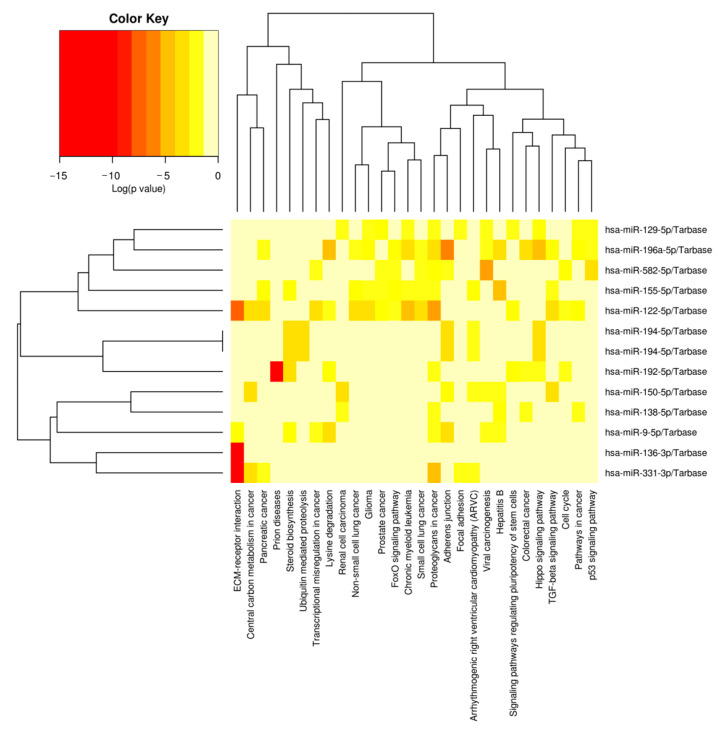
Cluster heat map of DEMs and enrichment pathways associated with target gene. The miRNAs enriched in the similar pathways were clustered together, and the pathways enriched with similar miRNAs were clustered together. Since porcine miRNAs were not available to the DIANA miRPath, the prediction used human miRNAs instead.

**Figure 8 genes-13-02406-f008:**
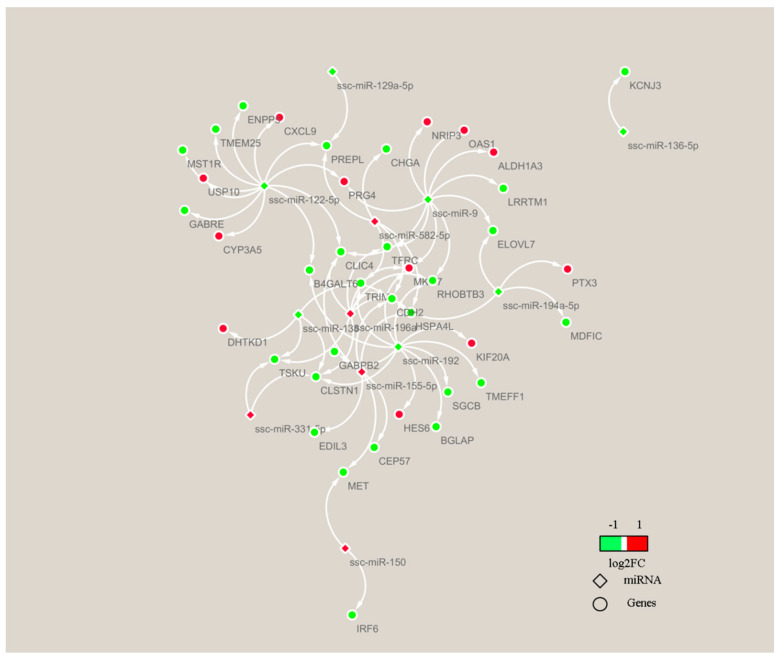
DEMs–target mRNA interaction network. The network of 11 DEMs and 39 target genes of miRNA were analyzed using Cytoscape.

**Table 1 genes-13-02406-t001:** Animal performance of Yorkshire pigs with FE-extreme individuals.

	High FE (*n* = 50)	Low FE (*n* = 50)	*p*-Value
FCR	2.23 ± 0.20	2.92 ± 0.31	9.60694 × 10^−24^
RFI (kg/day)	−0.28 ± 0.14	0.24 ± 0.13	2.228 × 10^−36^
FI	1.78 ± 0.29	2.35 ± 0.28	3.02393 × 10^−16^
ADG	0.80 ± 0.14	0.82 ± 0.14	0.67
Initial BW (kg)	38.87 ± 2.12	39.27 ± 2.94	0.37
Final BW (kg)	90.13 ± 3.41	90.84 ± 4.27	0.24
AMBW	21.98 ± 0.82	22.19 ± 0.62	0.45
ABF (mm)	18.68 ± 2.74	21.49 ± 2.23	1.0061 × 10^−5^
LMA (cm^2^)	45.75 ± 6.22	44.78 ± 6.87	0.46

FE, feed efficiency. FCR, feed conversion ratio. RFI, residual feed intake. FI, feed intake. ADG, average daily gain. BW, body weight. AMBW, average metabolic body weight. ABF, average of back fat thicknesses (mm) measured at three points between 6th and 7th ribs (6th–7th BF) and at the 10th rib (10th BF). LMA, loin muscle area (cm^2^) measured between the 10th and 11th. *p*-value as calculated by *t*-test.

**Table 2 genes-13-02406-t002:** Summary of miRNA-seq data.

Reads	High FE-1	High FE-2	High FE-3	Low FE-1	Low FE-2	Low FE-3
Total reads	20,197,804	12,835,372	20,920,937	26,772,081	17,184,327	23,376,133
Clean reads	18,424,875	10,611,213	18,687,418	22,321,137	15,274,616	20,018,461
Qualified%	0.912	0.827	0.893	0.834	0.889	0.856
mapped	4,251,042	2,045,141	4,265,705	6,004,627	5,288,460	4,190,707
unmapped	14,173,833	8,566,072	14,421,713	16,316,510	9,986,156	15,827,754
mapped%	0.231	0.193	0.228	0.269	0.346	0.209
unmapped%	0.769	0.807	0.772	0.731	0.654	0.791

**Table 3 genes-13-02406-t003:** A list of 14 differentially expressed miRNAs in adipose tissues between high-FE and low-FE pigs.

miRNA	Ref miRNA (Human)	Fold Change (High/Low)	*p*-Value	Mature Sequences
ssc-miR-122-5p	miR-122-5p	−3.50	1.64483 × 10^−5^	uggagugugacaaugguguuu
ssc-miR-192	miR-192-5p	−2.98	0.000328674	cugaccuaugaauugacagccag
ssc-miR-194a-5p	miR-194-5p	−2.76	0.000665162	uguaacagcaacuccaugugga
ssc-miR-10386		−3.18	0.000765051	gucguccucucccucccuccu
ssc-miR-155-5p	miR-155-5p	2.29	0.001718442	uuaaugcuaauugugauaggggu
ssc-miR-150	miR-150-5p	2.71	0.001962553	ucucccaacccuuguaccagug
ssc-miR-194b-5p	miR-194-5p	−2.43	0.00437337	uguaacagcgacuccaugugga
ssc-miR-582-5p	miR-582-5p	3.55	0.015997654	uacaguuguucaaccaguuacu
ssc-miR-331-5p	miR-331-3p	2.08	0.019987215	gccccugggccuauccuagaac
ssc-miR-136-5p	miR-136-3p	−3.32	0.020242907	caucaucgucucaaaugagucu
ssc-miR-129a-5p	miR-129-5p	−2.02	0.020370961	cuuuuugcggucugggcuugc
ssc-miR-9	miR-9-5p	−2.22	0.023102635	ucuuugguuaucuagcuguauga
ssc-miR-138	miR-138-5p	−2.27	0.039803139	agcugguguugugaaucaggccgu
ssc-miR-196a	miR-196a-5p	2.07	0.04389995	uagguaguuucauguuguuggg

## Data Availability

Data presented in this study are available at request from the corresponding author.

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
