# Peer review of "Identification of Differentially Expressed miRNAs in Porcine Adipose Tissues and Evaluation of Their Effects on Feed Efficiency"

_genes, 2022, doi:10.3390/genes13122406_

Round 1

Reviewer 1 Report

The manuscript by Liao et al. is well written and nicely described the results of differentially expressed miRNAs in porcine adipose tissues. My main suggestions are listed below:

1.     Figures 4 and 5 are all based on predicted target genes. Which version of mirPath was used? There are many predicted target genes. I wonder what approach you used to control for false positives.

2.     I think it’s worthwhile to expand the discussion of DE genes predicted as DEM targets.

3.     Since there are only 14 DEMs, why skip miR-331, miR-129, miR-9, and miR-196a?

Here are some minor suggestions:

Line 117-118: Miss the value for p-value cutoff. Did you use p-value or adjusted p-value? It’d be stricter to use adjusted p-values.

Line 173 and 174: Need to rewrite – the miRNAs were expressed higher, or the expression levels of the miRNAs were higher.

Figure 5: Is the X axis the number of miRNAs? Should it be the number of predicted target genes?

Figure 6: Not sure about what Figure 6 is trying to convey. Those functionally similar miRNAs were not further discussed in the text.

Author Response

Dear reviewers,

We appreciate all the suggestions and comments very much. All of the questions have been answered one by one. The manuscript has been revised carefully according to the suggestions of the reviewers. All the changes in the revised version were highlighted with bright yellow color.  

Many thanks to editors and reviewers for the efforts.                                                                                                                                           

Sincerely yours,

                                                                                                                                            Mingxing Liao, on behalf of all the authors                                                                                                                                              

Comments and Suggestions for Authors

The manuscript by Liao et al. is well written and nicely described the results of differentially expressed miRNAs in porcine adipose tissues. My main suggestions are listed below:

  1.     Figures 4 and 5 are all based on predicted target genes. Which version of mirPath was used? There are many predicted target genes. I wonder what approach you used to control for false positives.

AU:  DIANA miRPath (v.3) was used to predicted target genes, I have added the version number in this part (Line 156) . The high false positive rate is a difficult problem to solve, even the most advanced miRNA target prediction algorithms exhibit high false positive rates [1,2]. To control for false positives, the overlap of predicted target genes and differentially expressed genes in adipose tissues between high and low FE pigs were used for further analysis.

  1.     I think it’s worthwhile to expand the discussion ofDE genes predicted as DEM targets.

AU:  Thank you for your suggestion, we have added the discussion of DE genes predicted as DEM targets in the discussion section. (Line 356-368)

  1.     Since there are only 14 DEMs, why skip miR-331, miR-129, miR-9, and miR-196a?

AU:  Thank you for your suggestion, we have added the discussion of miR-331, miR-129, miR-9, and miR-196a in the discussion section. (Line324-327, Line 338-355 )

Here are some minor suggestions:

Line 117-118: Miss the value for p-value cutoff. Did you use p-value or adjusted p-value? It’d be stricter to use adjusted p-values.

-AU:Thank you for your suggestion, we have added the p-value cutoff in the section (Line 135-136). In this section, we used p-value instead of adjusted p-value, because the adjusted p-value are stricter and may filer out some important miRNAs.

Line 173 and 174: Need to rewrite – the miRNAs were expressed higher, or the expression levels of the miRNAs were higher.

-AU:Thank you for your suggestion, we have rewrite this part (Line 202-204). 

Figure 5: Is the X axis the number of miRNAs? Should it be the number of predicted target genes?

-AU:The X axis is the number of miRNAs, the number of predicted target genes is represented by the size of dot. In this part, DIANA miRPath used to explore the functions of significantly DE miRNAs between high-FE and low-FE pig, the detail information were list in Table S5.   

Figure 6: Not sure about what Figure 6 is trying to convey. Those functionally similar miRNAs were not further discussed in the text.

-AU:In Figure 6, the miRNAs enriched in the similar pathways were clustered together, and the pathways enriched with the similar miRNAs were clustered together. We have added the discussion of similar miRNAs in the text, for example miR-129-5p and miR-196a-5p, miR-150-5p and miR-138-5p. (Line 243-246, Line 341-349, Line 334-340).  

Reference

  1.  Vlachos I.S., Hatzigeorgiou A.G. Online resources for miRNA analysis. Clin. Biochem. 2013;46:879–900.

  1.  Vlachos IS, Zagganas K, Paraskevopoulou MD, Georgakilas G, Karagkouni D, Vergoulis T, Dalamagas T, Hatzigeorgiou AG. DIANA-miRPath v3.0: deciphering microRNA function with experimental support. Nucleic Acids Res. 2015 Jul 1;43(W1):W460-6. doi: 10.1093/nar/gkv403. Epub 2015 May 14. PMID: 25977294; PMCID: PMC4489228.

Reviewer 2 Report

This manuscript evaluated miRNA transcriptome in pig adipose tissues from high feed efficiency (FE) and low feed efficiency animals. The research started with a large sample set of 236 pigs that were evaluated for feed efficiency. From that, they selected 3 with low RFI (high FE) and 3 with high RFI for miRNA transcriptomic analyses of adipose tissues. Numbers used for miRNA sequencing are borderline but they were at least selected based on feed efficiency from a large sample set. There are a few issues with the lack of detail in the methods section that needed to be addressed. How was individual feed intake measured? An additional figure showing the variation in intake or RFI would be a nice addition so that we can see this and confirm that major differences in RFI were obtained. There is no description on what adipose tissue location was sampled. Was this subcutaneous fat at the 10th rib? If subcutaneous fat, which layer (outer, inner or middle) was used for RNA extraction and sequencing. No information was provided on how RNA was extracted and if checked for quality before sequencing. This may be in the companion paper that is reference but it is a critical detail that is missing in this manuscript. There are variable measures of feed efficiency (FE Feed:gaim; FCR gain:feed and RFI) discussed in this study and it would be cleared if the authors chose one, like RFI as this was how the low and high groups were defined and used it throughout. Overall the manuscript provides novel information on how miRNA in adipose tissues may be involved in feed efficiency but needs some additional detail and some edits.

Specific comments:

15 delete to after tissues and before regulate.

81 How was individual feed intake measured in this study? What was the overall variation in RFI or FE? A figure showing the differences in RFI for this study would be nice to see so that the reader can determine if the sample size was appropriate.

81 The use of 3 samples from high and low FE pigs is borderline for sequencing.

83 they establish process was consistent with previously published paper--this does not make sense and needs to be reworded.

90 It is critical that you describe what adipose tissue location was sampled for this study? Was it subcutaneous adipose tissue? If subcutaneous, which layer of fat was sampled? Where was it sampled from? This information must be detailed and provided in the manuscript.

Table 1. The table should be able to stand alone so all abbreviations should be defined and how they were calculated or where measured. FCR not defined, AMBW not defined, where was ABF measured, Where was LMA measured?

97 Methods on RNA extraction needs to be included. What kind of quality checks were run on the RNA before sequencing?

118 It does not appear that you used a false discovery rate (FDR) to control for multi-testing error in determining differentially expressed genes. This needs to be addressed.

119-126 Are these the same samples that were used for sequencing or different samples? More detail is needed here on methods including RT reaction and primers/qPCR methods. Did you use Taq Man probes? How were data statistically analyzed for qPCR.

221-226 There is a comment here about target genes that were differentially expressed in adipose tissues between the groups. There is no mention of mRNA qPCR methods used to evaluating targets. This must be included in methods section. Is this from the software program or qPCR results?

237 Please change to: ....tissue that take part in energy metabolism and is associated....

255 Please change to: .....pathway by mediating the level of.....

244 There should be discussion on the ABF levels between the High FE and low FE groups. This difference is small (2.81 mm) and only 13%. If these are major differences in FE and miRNA associated with adipose tissues involved, would we not expect greater differences in backfat? More discussion is needed here on this topic and may also be related to which location the fat samples were taken from for sequencing.

291 More discussion on how these miRNA may be involved in regulating feed efficiency is needed instead of just potential effects on adipose tissue metabolism.

293-297 Conclusions could be stronger to propose how these miRNA in adipose tissues might alter FE.

Author Response

Dear reviewers,

We appreciate all the suggestions and comments very much. All of the questions have been answered one by one. The manuscript has been revised carefully according to the suggestions of the reviewers. All the changes in the revised version were highlighted with bright yellow color.  

Many thanks to editors and reviewers for the efforts.

                                                                                                                                          Sincerely yours,

                                                                                                                                          Mingxing Liao, on behalf of all the authors                                                                                                                                             

Comments and Suggestions for Authors

This manuscript evaluated miRNA transcriptome in pig adipose tissues from high feed efficiency (FE) and low feed efficiency animals. The research started with a large sample set of 236 pigs that were evaluated for feed efficiency. From that, they selected 3 with low RFI (high FE) and 3 with high RFI for miRNA transcriptomic analyses of adipose tissues. Numbers used for miRNA sequencing are borderline but they were at least selected based on feed efficiency from a large sample set. There are a few issues with the lack of detail in the methods section that needed to be addressed. How was individual feed intake measured? An additional figure showing the variation in intake or RFI would be a nice addition so that we can see this and confirm that major differences in RFI were obtained. There is no description on what adipose tissue location was sampled. Was this subcutaneous fat at the 10th rib? If subcutaneous fat, which layer (outer, inner or middle) was used for RNA extraction and sequencing. No information was provided on how RNA was extracted and if checked for quality before sequencing. This may be in the companion paper that is reference but it is a critical detail that is missing in this manuscript. There are variable measures of feed efficiency (FE Feed:gaim; FCR gain:feed and RFI) discussed in this study and it would be cleared if the authors chose one, like RFI as this was how the low and high groups were defined and used it throughout. Overall the manuscript provides novel information on how miRNA in adipose tissues may be involved in feed efficiency but needs some additional detail and some edits.

AU: Thank you for your suggestion. We have added the detail description such as individual feed intake measured method, a figure showing the variation in intake or RFI, adipose tissue location, RNA extract method in the Materials and Methods section. All of the questions have been answered one by one in the follow.

Specific comments:

15 delete to after tissues and before regulate.

AU: Thank you for your suggestion, we have delete “to” in this part. (Line 15).

81 How was individual feed intake measured in this study? What was the overall variation in RFI or FE? A figure showing the differences in RFI for this study would be nice to see so that the reader can determine if the sample size was appropriate.

AU: Thank you for your suggestion, we added some detailed information about individual feed intake measured in the material and methods section. (Line 84-85)

The overall variation in RFI is -0.03188±0.2051, and the figure showing the differences in RFI was added in Figure 1.(Line 86-87, Line101-104)

81 The use of 3 samples from high and low FE pigs is borderline for sequencing.

AU: A large number of biological duplications can improve the accuracy of identifying gene expression levels and discover meaningful miRNAs. However, due to slaughtering costs and sample limitations, it is often difficult to have a large enough sample size to carry out the experiment. For domestic animals, many literatures indicated that 6 individuals (3 vs. 3) were sufficient to transcriptome analysis. Therefore, we slaughtered 6 pigs and obtained 6 samples for sequencing in our experiment. Although the sample size is relatively small, I think some meaningful miRNAs can still be found. In the future experiments, we will pay attention to the influence of the number of experimental individuals on the experiment and increase the number of experimental individuals as much as possible.

83 they establish process was consistent with previously published paper--this does not make sense and needs to be reworded.

AU: Thank you for your suggestion, we have revised this part and added the information about  individual feed intake measured. (Line 84-85)

90 It is critical that you describe what adipose tissue location was sampled for this study? Was it subcutaneous adipose tissue? If subcutaneous, which layer of fat was sampled? Where was it sampled from? This information must be detailed and provided in the manuscript.

AU:Thank you for your suggestion, we have added the detailed information about the adipose tissue in the sample preparation and RNA isolation section. (Line 93-95)

Table 1. The table should be able to stand alone so all abbreviations should be defined and how they were calculated or where measured. FCR not defined, AMBW not defined, where was ABF measured, Where was LMA measured?

AU: Thank you for your suggestion, we have added the defined of FCR, AMBW, RFI and we have added the measured for ABF and LMA (Line 107-110).   

97 Methods on RNA extraction needs to be included. What kind of quality checks were run on the RNA before sequencing?

AU: Thank you for your suggestion, we have added the methods on RNA extraction and the information about quality control (Line 96-100).

118 It does not appear that you used a false discovery rate (FDR) to control for multi-testing error in determining differentially expressed genes. This needs to be addressed.

-AU:It has been reported that p-value <0.05 standard used to detected DE miRNAs. [1-4] Previous studies also used P-values to select differentially expressed miRNAs, and the miRNAs we focused on were miRNAs with the lowest P-value, which also had statistical significance. FDR reduces false positives, but it also increases false negatives.

119-126 Are these the same samples that were used for sequencing or different samples? More detail is needed here on methods including RT reaction and primers/qPCR methods. Did you use Taq Man probes? How were data statistically analyzed for qPCR.

AU: The samples used for qRT-PCR are the same samples. On the basis of sequencing (3 vs 3 ), we added two samples each group (5 vs 5) for verification. The detail description about RT reaction and primers/qPCR methods were added. In this study, we used SYBR Green PCR instead of Taq Man probes method. The relative expression levels of miRNAs were calculated using the 2−ΔΔCt method and the expression difference between the high-FE and low-FE pigs were analyze with the Student’s t-test. (Line 141-153)

221-226 There is a comment here about target genes that were differentially expressed in adipose tissues between the groups. There is no mention of mRNA qPCR methods used to evaluating targets. This must be included in methods section. Is this from the software program or qPCR results?

AU: The target genes that were differentially expressed in adipose tissues between the groups was reported in our previous paper [5]. In our previous paper, mRNA sequencing was ued to identify the differentially expressed in adipose tissues and mRNA qPCR methods was used to validated the reliability of mRNA sequencing . The overlap between differentially expressed genes (high FE vs. low FE adipose tissues) and 5927 possible target genes were considered as the target genes that were differentially expressed in adipose tissues. We have added this part in the paper. (Line 160-162)

237 Please change to: ....tissue that take part in energy metabolism and is associated....

AU: Thank you for your suggestion, we have revised the description of this part. (Line 268-269).            

255 Please change to: .....pathway by mediating the level of.....

AU: Thank you for your suggestion, we have revised this part. (Line 295-297).

244 There should be discussion on the ABF levels between the High FE and low FE groups. This difference is small (2.81 mm) and only 13%. If these are major differences in FE and miRNA associated with adipose tissues involved, would we not expect greater differences in backfat? More discussion is needed here on this topic and may also be related to which location the fat samples were taken from for sequencing.

AU: Thank you for your suggestion, we have added the discussion on the ABF levels between the High FE and low FE groups. Feed efficiency is a complex trait, several tissues such as muscle, liver, duodenum, hypothalamus, and adipose has been report to be associated with the FE of pigs.  Energy metabolism is the main factor affecting the feed efficiency of pigs, and adipose tissue is an important tissues involved in energy metabolism [6-9]. Moreover, the adipose tissue can secrete many adipocytokines to mediate appetite, and thus regulate feed efficiency by influencing feed intake [10]. Therefore the adipose tissue acts as an energy depot to mediate metabolic homeostasis and nutrient availability,thereby regulating feed efficiency. (Line 273-284).

291 More discussion on how these miRNA may be involved in regulating feed efficiency is needed instead of just potential effects on adipose tissue metabolism.

AU: Thank you for your suggestion, we have added the discussion on these miRNA involved in regulating feed efficiency . (Line 277-284, Line 354-355)

293-297 Conclusions could be stronger to propose how these miRNA in adipose tissues might alter FE.

AU: Thank you for your suggestion, we have added the content in conclusions to revea how these miRNA in adipose tissues might alter FE . ( Line 373-376)

Reference

  1. Zhang B, Qiangba Y, Shang P, Wang Z, Ma J, Wang L, Zhang H: A Comprehensive MicroRNA Expression Profile Related to Hypoxia Adaptation in the Tibetan Pig. PloS one 2015, 10(11):e0143260.
  2. Xu Y, Qi X, Hu M, Lin R, Hou Y, Wang Z, Zhou H, Zhao Y, Luan Y, Zhao S et al: Transcriptome Analysis of Adipose Tissue Indicates That the cAMP Signaling Pathway Affects the Feed Efficiency of Pigs. Genes 2018, 9(7).
  3. Khatri B, Seo D, Shouse S, Pan JH, Hudson NJ, Kim JK, Bottje W, Kong BC: MicroRNA profiling associated with muscle growth in modern broilers compared to an unselected chicken breed. BMC genomics 2018, 19(1):683.
  4. Wang D, Xin L, Lin JH, Liao Z, Ji JT, Du TT, Jiang F, Li ZS, Hu LH: Identifying miRNA-mRNA regulation network of chronic pancreatitis based on the significant functional expression. Medicine 2017, 96(21):e6668.
  5.  Xu Y, Qi X, Hu M, Lin R, Hou Y, Wang Z, et al. Transcriptome Analysis of Adipose Tissue Indicates That the cAMP Signaling Pathway Affects the Feed Efficiency of Pigs. Genes. 2018;9(7).
  6.  Herd RM, Arthur PF. Physiological basis for residual feed intake. J Anim Sci. 2009 Apr;87(14 Suppl):E64-71. doi: 10.2527/jas.2008-1345. Epub 2008 Nov 21. PMID: 19028857.
  7.  Horodyska, J,Oster, M,Reyer, H, et al. Analysis of meat quality traits and gene expression profiling of pigs divergent in residual feed intake. Meat Sci. 2018, 137, 265–274.
  8.  Ramayo-Caldas, Y,Ballester, M,Sanchez, J.P, et al. Integrative approach using liver and duodenum RNA-Seq data identifies candidate genes and pathways associated with feed efficiency in pigs. Sci. Rep. 2018, 8, 558.
  9.   Ding, R,Quan, J,Yang, M, et al. Genome-wide association analysis reveals genetic loci and candidate genes for feeding behavior and eating efficiency in Duroc boars. PLoS ONE 2017, 12.
  10.  Faust IM, Johnson PR, Hirsch J. Surgical removal of adipose tissue alters feeding behavior and the development of obesity in rats. Science. 1977;197(4301):393-6.

Round 2

Reviewer 2 Report

The authors have addressed my comments and improved the manuscript.